# Anti-inflammatory therapy with nebulized dornase alfa for severe COVID-19 pneumonia: a randomized unblinded trial

Joanna C Porter[1,2]*, Jamie Inshaw[3], Vincente Joel Solis[2], Emma Denneny[1,2], Rebecca Evans[2], Mia I Temkin[4†], Nathalia De Vasconcelos[4†], Iker Valle Aramburu[4†], Dennis Hoving[4†], Donna Basire[1], Tracey Crissell[2], Jesusa Guinto[2], Alison Webb[2], Hanif Esmail[2,5], Victoria Johnston[2,5], Anna Last[2,6], Thomas Rampling[2,5], Lena Lippert[7], Elisa Theresa Helbig[7], Florian Kurth[7], Bryan Williams[2,5], Aiden Flynn[3], Pauline T Lukey[8], Veronique Birault[9], Venizelos Papayannopoulos[4]*

[1]UCL Respiratory, University College London, London, United Kingdom; [2]University College London Hospitals NHS Trust, London, United Kingdom; [3]Exploristics, Belfast, Ireland; [4]Antimicrobial Defence Lab, The Francis Crick Institute, London, United Kingdom; [5]National Institute for Health Research, University College London Hospital Biomedical Research Centre, London, United Kingdom; [6]Clinical Research Department, London School of Hygiene and Tropical Medicine, London, United Kingdom; [7]Charité – Universitätsmedizin Berlin, Department of Infectious Diseases and Respiratory Medicine, Berlin, Germany; [8]Target to Treatment Consulting Ltd, Stevenage, United Kingdom; [9]Translation, The Francis Crick Institute, London, United Kingdom

*For correspondence:
joanna.porter@ucl.ac.uk (JCP);
veni.p@crick.ac.uk (VP)

†These authors contributed equally to this work

## Abstract

**Background:** Prinflammatory extracellular chromatin from neutrophil extracellular traps (NETs) and other cellular sources is found in COVID-19 patients and may promote pathology. We determined whether pulmonary administration of the endonuclease dornase alfa reduced systemic inflammation by clearing extracellular chromatin.

**Methods:** Eligible patients were randomized (3:1) to the best available care including dexamethasone (R-BAC) or to BAC with twice-daily nebulized dornase alfa (R-BAC + DA) for seven days or until discharge. A 2:1 ratio of matched contemporary controls (CC-BAC) provided additional comparators. The primary endpoint was the improvement in C-reactive protein (CRP) over time, analyzed using a repeated-measures mixed model, adjusted for baseline factors.

**Results:** We recruited 39 evaluable participants: 30 randomized to dornase alfa (R-BAC +DA), 9 randomized to BAC (R-BAC), and included 60 CC-BAC participants. Dornase alfa was well tolerated and reduced CRP by 33% compared to the combined BAC groups (T-BAC). Least squares (LS) mean post-dexamethasone CRP fell from 101.9 mg/L to 23.23 mg/L in R-BAC +DA participants versus a 99.5 mg/L to 34.82 mg/L reduction in the T-BAC group at 7 days; p=0.01. The anti-inflammatory effect of dornase alfa was further confirmed with subgroup and sensitivity analyses on randomised participants only, mitigating potential biases associated with the use of CC-BAC participants. Dornase alfa increased live discharge rates by 63% (HR 1.63, 95% CI 1.01–2.61, p=0.03), increased lymphocyte counts (LS mean: 1.08 vs 0.87, p=0.02) and reduced circulating cf-DNA and the coagulopathy marker D-dimer (LS mean: 570.78 vs 1656.96 µg/mL, p=0.004).

**Conclusions:** Dornase alfa reduces pathogenic inflammation in COVID-19 pneumonia, demonstrating the benefit of cost-effective therapies that target extracellular chromatin.

**Funding:** LifeArc, Breathing Matters, The Francis Crick Institute (CRUK, Medical Research Council, Wellcome Trust).
**Clinical trial number:** NCT04359654.

## eLife assessment

This small-sized clinical trial comparing nebulized dornase-alfa to the best available care in patients hospitalized with COVID-19 pneumonia is **valuable**, but in its present form the paper is **incomplete**.

## Introduction

SARS-CoV-2 pneumonia can lead to hyperinflammation, coagulopathy, respiratory failure, and death (*Siddiqi and Mehra, 2020*; *Zhou et al., 2020*), even with less pathogenic variants and in immunized patients (*American Thoracic Society, 2000*). In severe COVID-19 pneumonia, neutrophil activation drives neutrophil extracellular trap (NET) formation (*Radermecker et al., 2020*; *Zuo et al., 2020*). NETs, are composed of DNA-histone complexes that are associated with coagulopathy and endothelial dysfunction (*Papayannopoulos, 2018*). Extracellular histones from NETs, or other cellular sources such as dying lymphocytes, promote inflammation, immune dysfunction, and lethality in sepsis (*Ioannou et al., 2022*; *Tsourouktsoglou et al., 2020*; *Xu et al., 2011*; *Xu et al., 2009*). Digestion of chromatin DNA by endonucleases suppresses the proinflammatory activity of histones and enables their clearance from the circulation (*Ioannou et al., 2022*; *Tsourouktsoglou et al., 2020*). Consistently, DNAse I treatment reduces pathology in murine pulmonary viral infections (*Cortjens et al., 2018*; *Pillai et al., 2016*). Endogenous DNAse activity and NET clearance capacity are defective in severe COVID-19 pneumonia, and the extent of these defects correlates with mortality (*Aramburu et al., 2022*). Hence, supplementation with exogenous DNases may facilitate extracellular chromatin degradation to reduce pathology and increase survival.

Dornase alfa, the active ingredient in pulmozyme, is a recombinant human DNase approved since 1993 for patients with cystic fibrosis (CF) (*Konstan and Ratjen, 2012*; *Lazarus and Wagener, 2019*). Dornase alfa solubilizes NETs, reduces inflammation, and improves pulmonary function in chronic and acute exacerbations of CF (*Konstan and Ratjen, 2012*; *Papayannopoulos et al., 2011*). Pulmozyme is safe and well-tolerated in children and adults with CF at doses up to 10 mg taken twice daily. Since nebulized pulmozyme administration does not increase circulating endonucleases, it remains unclear whether the degradation of extracellular chromatin in the lungs would reduce circulating chromatin and systemic hyper-inflammation in viral infections.

To probe the effectiveness of DNase I as an anti-inflammatory therapy in severe COVID-19 pneumonia, we measured as our primary endpoint C-reactive protein (CRP). CRP is a reliable marker of systemic inflammation in severe infection (*McArdle et al., 2004*). In patients infected with COVID-19, a high CRP of ≥40 mg/L carried a poor prognosis with a lower cut-off of ≥35 mg/L in the elderly (*Villoteau et al., 2021*), particularly when combined with lymphopenia and coagulopathy marked by elevated D-dimer (*Fisher et al., 2022*; *Liu et al., 2020*; *Luo et al., 2020*; *Smilowitz et al., 2021*; *Tornling et al., 2021*; *Ullah et al., 2020*). Poor outcomes are associated with an early rise (*Mueller et al., 2020*), higher peak, and delayed reduction in CRP (*Cui et al., 2021*). Moreover, elevated IL-6 and CRP can predict the need for mechanical ventilation (*Herold et al., 2020*). In addition, CRP correlated well with levels of DNA and NET degradation activity in COVID-19 patient plasma suggesting a link between CRP and the capacity to degrade cell free (cf)-DNA in the circulation (*Aramburu et al., 2022*). CRP became a reliable primary endpoint in subsequent trials that demonstrated the effectiveness of other systemic therapies such as namilumab or infliximab that target inflammatory cytokines (*Fisher et al., 2022*).

In this proof-of-concept study, we evaluated the effect of dornase alfa on inflammation, as measured by the impact on circulating CRP in patients with COVID-19, compared with best available care (BAC). The trial was initiated in June 2020 and was completed in October 2021. At the start of the trial, only dexamethasone had been proven to benefit hospitalized COVID-19 pneumonia patients and was thus included in both arms of the trial. To increase the chance of reaching significance under constraints in patient access, we opted to increase our sample size by using a combination of randomized individuals in the BAC group (R-BAC) and the dornase alfa treated arm

(R-BAC +DA) with available CRP data from matched contemporary controls (CC-BAC) who were hospitalized at the same site at University College London Hospital (UCLH) but were not recruited to a trial. The same selection criteria were applied to CC-BAC control participants as in the randomized group to minimize potential biases. In the analysis framework, we compared treated R-BAC-DA participants to R-BAC or CC-BAC groups individually or by combining them into a total BAC group (T-BAC). These approaches demonstrated that when combined with dexamethasone, nebulized DNase treatment was an effective anti-inflammatory treatment in randomized individuals with or without the implementation of CC-BAC data.

## Methods

**Key resources table**

| Reagent type (species) or resource | Designation | Source or reference | Identifiers | Additional information |
|---|---|---|---|---|
| Recombinant DNA reagent | $\lambda$ DNA (clind 1 ts857 Sam 7) | Invitrogen | 25250010 | |
| Commercial assay or kit | Quant-iT PicoGreen dsDNA Reagent | Invitrogen | P7581 | |
| Chemical compound, drug | Histopaque-1119 | Sigma-Aldrich | 11191 | Density media for cell isolation |
| Chemical compound, drug | Percoll | Cytiva | 17544502 | Density media for cell isolation |
| Chemical compound, drug | HBSS without calcium, magnesium, phenol red | Cytiva | SH30588.01 | Balanced salt solution |
| Chemical compound, drug | 1 M HEPES | Sigma-Aldrich | SRE0065 | Biological buffer |
| Chemical compound, drug | Tris-EDTA | Sigma-Aldrich | 93283 | Biological buffer |

### Sponsor and location

The trial was sponsored by University College London (UCL) and carried out at UCLH with ethical (REC: 20/SC/0197, Protocol: 132333, RAS ID:283091) and UK MHRA approvals. All randomized participants provided informed consent. Consent for CC-BAC participants was covered by Health Service (Control of Participant Information) Regulations 2002. Safety and data integrity were overseen by the Trial Monitoring Group and Data Monitoring Committee. All data were collected at UCLH. Additional CRP data is reported from the Pa-COVID-19 study undertaken at the Charité - Universitätsmedizin Berlin with ethical approval, Berlin (EA2/066/20). Both Pa-COVID-19 and COVASE studies were carried out according to the Declaration of Helsinki and the principles of Good Clinical Practice (ICH 1996).

### Trial design

The COVASE trial was a single-site, randomized, controlled, parallel, open-label investigation. The primary endpoint was change in CRP. Screening was performed within 24 hr prior to the administration of dornase alfa (*Figure 1A*). Eligible, consented participants were randomly assigned (3:1) to either the BAC plus nebulized dornase alfa (R-BAC +DA) or the R-BAC group alone. On day 1, a baseline sample was collected. From day 1–7, participants that were randomized to the active arm received twice daily 2.5 mg nebulized dornase alfa in addition to BAC. In all cases BAC included dexamethasone (6 mg/day) for 10 days or until discharge, whichever was shorter as established in the RECOVERY trial (*Horby et al., 2021*). Participants received additional treatments at the discretion of their physicians. The primary analysis was performed on samples up to day 7. The final trial visit occurred on day 35.

In addition to the recruited randomized group, we included CC-BAC participants to increase the sample size due to practical difficulties in recruiting patients in the UK. For every COVASE participant randomized to the active treatment, two matched CC-BAC participants were included. CC-BAC participants were admitted to UCLH over the same time as randomized patients and treated with the same BAC regimen, including dexamethasone. CC-BAC participants fulfilled the inclusion and exclusion criteria of the COVASE study and were matched for age, gender, BMI, comorbidities and CRP. CRP matching was based either on admission CRP, or matched based on the first CRP reading after starting treatment with dexamethasone.

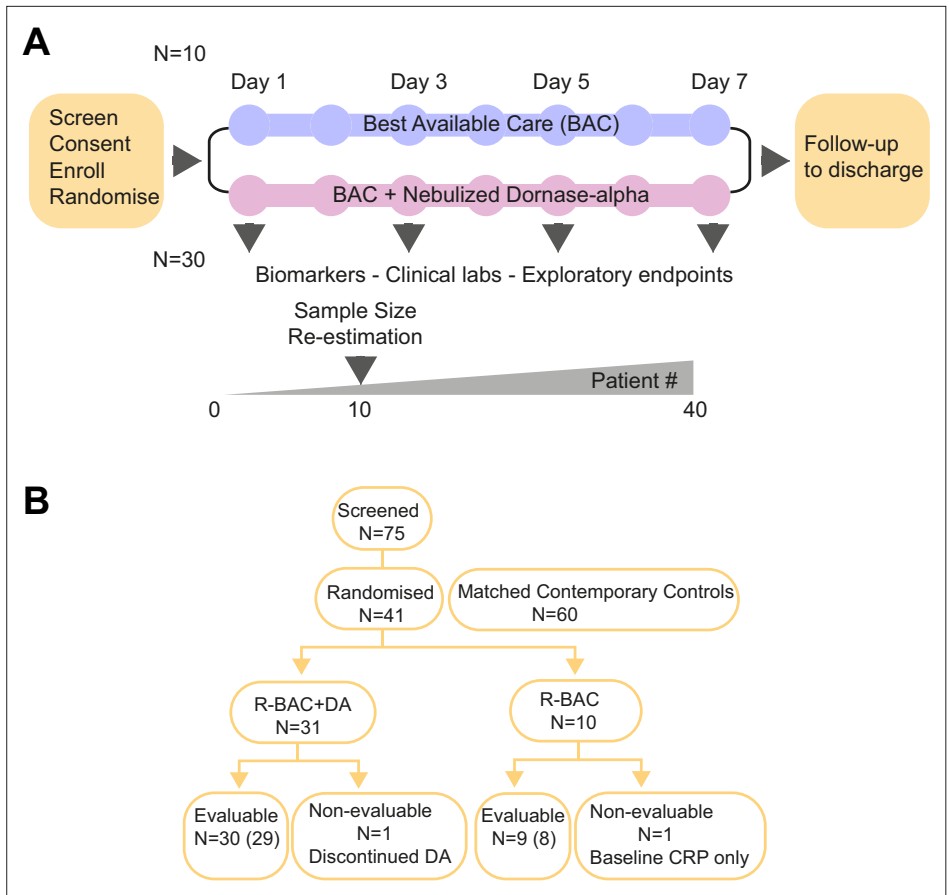

**Figure 1.** Trial design and Consort diagram. (**A**) COVASE Trial Design. (**B**) Consort diagram summary. Numbers not in parentheses indicate the participants in the intention-to-treat (ITT) population and the numbers in parentheses indicate the number of participants in the per-protocol population. A complete consort flow diagram is shown in *Figure 1—figure supplement 1*.

The online version of this article includes the following source data and figure supplement(s) for figure 1:

**Figure supplement 1.** Consort flow diagram.

**Figure supplement 2.** Baseline characteristics of patients analysed in the trial.

**Figure supplement 2—source data 1.** Baseline characteristics in T-BAC and R-BAC +DA participants.

## Eligibility
### Trial participant inclusion criteria
Male and female participants, aged ≥18 years, who were hospitalized for suspected Coronavirus 2 (SARS-CoV–2) infection confirmed by polymerase chain reaction (PCR) test and optionally with radiological confirmation with chest CT. Participants with stable oxygen saturation (≥94%) on supplementary oxygen and a CRP ≥30 mg/L. Participants that provided written informed consent to participate in the study and were able to comply with instructions and the use of a nebulizer.

### Exclusion criteria
Females who were pregnant, planning pregnancy, or breastfeeding. Concurrent and/or recent involvement in other research or use of another experimental investigational medicinal product that was likely to interfere with dornase alfa within 3 months of study enrolment. Serious condition meeting either respiratory distress with respiratory rate ≥40 breaths/min, or an oxygen saturation ≤93% on high-flow oxygen. Participants who required mechanical invasive or non-invasive ventilation at screening or had concurrent severe respiratory disease such as asthma, chronic obstructive pulmonary disease (COPD) and/or interstitial lung disease (ILD), or any major disorder that in the opinion of the Investigator

would interfere with the evaluation of the results or constitute a health risk for the trial participant. Participants with terminal disease and life expectancy <12 months without COVID-19, or with known allergies to dornase alfa and excipients, or participants who were unable to inhale or exhale orally throughout the entire nebulization period.

## Participants

Adults (≥18 years of age) admitted to UCLH with confirmed SARS-Cov2 infection by RT-PCR and radiologically confirmed COVID-19 pneumonia on chest radiograph or CT-scan; oxygen saturation <94% requiring supplemental oxygen; and evidence of hyperinflammation (CRP ≥30 mg/L, after administration of dexamethasone) were eligible. While all participants had CRP >30 mg/L at screening, on two occasions the baseline CRP concentrations fell below 30 mg/L after the patients had already agreed to participate. Full inclusion and exclusion criteria are provided in the study protocol (*Supplementary file 2*).

## Outcomes

The primary outcome was the least square (LS) mean CRP concentration up to 7 days or at hospital discharge whichever was sooner. Data beyond 7 days were included in the model to estimate the slope. Subsequently, the estimated slope in each group was used in order to calculate the LS mean at 7 days. Pre-specified secondary outcomes included days on oxygen; time to hospital discharge; mortality by day 35; and changes in clinically relevant biomarkers including lymphocyte count and D-dimer levels. Day 35 was chosen as being likely to include early mortality due to COVID-19 being 4 weeks after completion of a week of treatment.

Efficacy assessments of primary and secondary outcomes in the intention-to-treat (ITT) population were performed on all randomized participants who had received at least one dose of dornase alfa, if randomized to treatment. The ITT was adjusted to mitigate the following protocol violations where one participant in the R-BAC arm and one in the R-BAC +DA arm withdrew before they received treatment and provided only a baseline CRP measurement available. The participant in the BAC +DA arm was replaced with an additional recruited participant. Exploratory endpoints were only available in randomized participants and not in the CC-BAC group. In this case, a *post hoc* within group analysis was conducted to compare baseline and post-baseline measurements. The CONSORT checklist is provided in *Supplementary file 2*.

## Sample size calculation

Size calculations were produced using the proc power function in the statistical analysis plan (SAP) (*Supplementary file 3*). These were conducted to achieve 80% power to detect a difference in the active arm versus the control group at a 5% level of significance. Based on a mean of 99 mg/L in the control group and a common standard deviation of 62 mg/L derived from the literature (*Han et al., 2020*; *Zhou et al., 2020*), a total sample size of 90 participants would provide sufficient power to detect a greater than a 40% relative difference in CRP between the dornase alfa treated group and the combined randomised and contemporary control groups.

This study used existing data collected at UCLH from CC-BAC participants admitted with COVID-19. This gave a ratio of active treatment versus comparator of 1:2. The required power, would result in 30 participants in the active treatment group and at least 60 participants in the control group. An additional 10 randomized control participants were recruited for exploratory endpoints and to compare the characteristics of enrolled participants to CC-BAC participants. Therefore, a total of 40 randomised participants were enrolled in the study with the addition of 60 CC-BAC participants.

## Randomization

A closed-envelope method was implemented to randomize the participants into the control and active treatment groups.

# Results

## Patient characteristics

From June 2020 to October 2021, 41 participants were recruited and randomized, with 1 participant in R-BAC group discharged before a second CRP measurement who was excluded from all analyses except for safety (CONSORT Diagram: *Figure 1B* and *Figure 1—figure supplement 1*). One participant withdrew consent prior to receiving dornase alfa and was replaced and excluded from all analyses. Therefore, 39 participants were included in the ITT analysis set, 30 R-BAC +DA and 9 R-BAC. One participant withdrew due to side effects. This participant was removed from the per protocol population (PPP). All 39 participants were followed up for 35 days or death whichever was sooner. Two participants were excluded from the PPP and one from the R-BAC group as this was the only patient in whom randomization occurred prior to dexamethasone being widely used in COVID-19 treatment and a second participant who withdrew after one dose of dornase alfa (*Figure 1B* and *Figure 1—figure supplement 1*). The trial ended when 40 eligible participants had been recruited, although thereafter, one participant in the R-BAC arm had to be excluded from the primary and secondary endpoint analysis.

Baseline characteristics were well balanced across groups (*Table 1*). The selection of 60 matched CC-BAC participants via propensity score matching was successful in ensuring that the means of the characteristics included in the propensity score matching process were similar (*Table 1* and *Figure 1—figure supplement 2*). The overall mean age was 56.8 years (56.8 years R-BAC +DA, 56.8 years T-BAC). The percentage of males was 75.8% (76.7% R-BAC +DA, 75.4% T-BAC). The most prevalent ethnicity was 'White British,' with 30.3% of participants identifying in that category (33.3% BAC +DA, 29.0% T-BAC). The mean BMI was 28.0 kg/m$^2$ (27.8 kg/m$^2$ R-BAC +DA, 28.2 kg/m$^2$ T-BAC). The mean baseline CRP post dexamethasone was 100.2 mg/L (101.9 mg/L R-BAC +DA, 99.5 mg/L T-BAC). The proportion of participants with a comorbidity, defined as one or more of hypertension, diabetes, or cardiovascular disease, was 52.5% (46.7% R-BAC +DA, 55.1% T-BAC). All randomized participants, except one, received dexamethasone prior to randomization. In addition, 48 of the total 99 participants also received remdesivir or tocilizumab in addition to dexamethasone within the first 7 days. The average duration of dexamethasone treatment prior to dornase alfa was 1.13+/-0.79 days (*Supplementary file 1A*). The last pre-dexamethasone CRP was also similar between groups with a mean of 125.0 mg/L (128.1 mg/L R-BAC +DA, 122.7 mg/L T-BAC). The number of days between dexamethasone initiation and baseline was 1.2 days (0.7 days R-BAC +DA, 1.3 days T-BAC). Imbalances were noted at baseline between the groups in white blood cell and neutrophil counts and procalcitonin and D-dimer concentrations (*Table 1*).

## Clinical outcomes

### Longitudinal CRP correlates with disease outcomes

To ascertain the clinical value of CRP as the COVASE trial's primary endpoint, we first examined the correlation of CRP with mortality in a cohort of 63 hospitalized patients with severe COVID-19 pneumonia (*Aramburu et al., 2022*; *Kurth et al., 2020*; *Messner et al., 2020*). All participants who reached a World Health Organisation (WHO) severity ordinance scale of 7 were recruited at the Charité Hospital in Berlin in the spring of 2020 and did not receive anti-inflammatory therapies such as dexamethasone or anti-IL-6 antibody treatments. Plasma CRP measurements in 465 samples were segregated by disease outcome. This analysis revealed that deceased participants had significantly higher plasma CRP concentrations than the participants who survived (*Figure 2A*). Clustering the 63 participants according to their longitudinal average CRP concentration indicated that the frequency of mortality increased proportionally to the average CRP concentration per patient (*Figure 2B*). Consistently, Mantel Cox survival analysis of the participants segregated into three groups with average longitudinal CRP concentration ranges of 0–100 mg/L, 100–200 mg/L, and 200–450 mg/L, showed that survival decreased significantly as the average CRP concentration increased (*Figure 2C*). These results highlighted the strong correlation between plasma CRP and the probability of survival in hospitalized individuals with severe COVID-19 pneumonia, suggesting that the blood CRP concentration was a strong predictor of clinical outcomes.

### Primary outcome

Next, we analyzed the results from the COVASE participants. The individual CRP traces over time are shown in *Figure 3—figure supplement 1*. Blood collection for both R-BAC and R-BAC +DA

**Table 1.** Patient baseline characteristics.

Age, gender, BMI, comorbidity frequency, CRP, WHO ordinal severity score, blood leukocyte counts, and blood procalcitonin and D-dimer concentrations at baseline for all participants (R-BAC, CC-BAC, and R-BAC +DA).

| | R-BAC +DA (n=30) | R-BAC (n=9) | CC-BAC (n=60) | T-BAC (n=69) | Total (n=99) |
|---|---|---|---|---|---|
| **Age (years)** | | | | | |
| Mean | 56.8 | 53.3 | 57.3 | 56.8 | 56.8 |
| SD | 12.5 | 13.7 | 14.5 | 14.3 | 13.7 |
| Median | 58.0 | 53.0 | 57.0 | 57.0 | 57.0 |
| Min | 32.0 | 31.0 | 23.0 | 23.0 | 23.0 |
| Max | 77.0 | 76.0 | 86.0 | 86.0 | 86.0 |
| **Gender** | | | | | |
| Female N (%) | 7 (23.3) | 2 (22.2) | 15 (25.0) | 17 (24.6) | 24 (24.2) |
| Male N (%) | 23 (76.7) | 7 (77.8) | 45 (75.0) | 52 (75.4) | 75 (75.8) |
| **BMI (kg/m$^2$)** | | | | | |
| Mean | 27.8 | 30.8 | 27.8 | 28.2 | 28.0 |
| SD | 4.7 | 7.8 | 5.6 | 6.0 | 5.6 |
| Median | 26.5 | 28.9 | 27.9 | 28.2 | 27.7 |
| Min | 20.7 | 22.6 | 16.3 | 16.3 | 16.3 |
| Max | 41.7 | 48.4 | 43.8 | 48.4 | 48.4 |
| **Baseline CRP (mg/L)** | | | | | |
| Mean | 101.9 | 91.9 | 100.7 | 99.5 | 100.2 |
| SD | 52.2 | 68.1 | 68.3 | 67.8 | 63.3 |
| Median | 86.3 | 74.6 | 75.8 | 75.3 | 79.6 |
| Min | 25.2 | 18.9 | 30.8 | 18.9 | 18.9 |
| Max | 261.5 | 221.6 | 336.4 | 336.4 | 336.4 |
| **Comorbidity** | | | | | |
| No N (%) | 16 (53.3) | 3 (33.3) | 28 (46.7) | 31 (44.9) | 47 (47.5) |
| Yes N (%) | 14 (46.7) | 6 (66.7) | 32 (53.3) | 38 (55.1) | 52 (52.5) |
| **WHO ordinal severity score** | | | | | |
| Mean | 5.0 | 5.0 | 4.63 | - | - |
| SD | 0.0 | 0.5 | 1.33 | - | - |
| Median | 5.0 | 5.0 | 5 | - | - |
| Min | 5.0 | 4.0 | 3 | - | - |
| Max | 5.0 | 6.0 | 7 | - | - |
| **WBC count (×10$^9$ /L)** | | | | | |
| N | 30 | 9 | 60 | 69 | 99 |
| Mean | 6.7 | 7.0 | 10.6 | 10.2 | 9.1 |
| SD | 2.5 | 2.7 | 9.2 | 8.7 | 7.6 |
| Median | 6.5 | 7.0 | 9.5 | 8.9 | 7.9 |
| Min | 3.1 | 1.8 | 1.8 | 1.8 | 1.8 |

*Table 1 continued on next page*

*Table 1 continued*

| | R-BAC +DA (n=30) | R-BAC (n=9) | CC-BAC (n=60) | T-BAC (n=69) | Total (n=99) |
|---|---|---|---|---|---|
| Max | 12.9 | 10.3 | 72.6 | 72.6 | 72.6 |
| **Neutrophil count (×10⁹ /L)** | | | | | |
| N | 30 | 9 | 60 | 69 | 99 |
| Mean | 5.7 | 5.6 | 9.1 | 8.7 | 7.8 |
| SD | 2.3 | 2.6 | 8.8 | 8.4 | 7.2 |
| Median | 5.3 | 5.8 | 7.9 | 7.9 | 6.7 |
| Min | 2.4 | 1.2 | 1.2 | 1.2 | 1.2 |
| Max | 10.9 | 8.6 | 69.5 | 69.5 | 69.5 |
| **Lymphocyte count (×10⁹ /L)** | | | | | |
| N | 30 | 9 | 60 | 69 | 99 |
| Mean | 0.7 | 0.9 | 0.9 | 0.9 | 0.9 |
| SD | 0.3 | 0.4 | 0.5 | 0.5 | 0.5 |
| Median | 0.5 | 0.9 | 0.8 | 0.8 | 0.7 |
| Min | 0.2 | 0.4 | 0.1 | 0.1 | 0.1 |
| Max | 1.5 | 1.5 | 3.7 | 3.7 | 3.7 |
| **Monocyte count (×10⁹ /L)** | | | | | |
| N | 30 | 9 | 60 | 69 | 99 |
| Mean | 0.4 | 0.4 | 0.5 | 0.5 | 0.4 |
| SD | 0.2 | 0.3 | 0.3 | 0.3 | 0.3 |
| Median | 0.3 | 0.3 | 0.4 | 0.4 | 0.4 |
| Min | 0.1 | 0.1 | 0.1 | 0.1 | 0.1 |
| Max | 0.9 | 0.8 | 1.7 | 1.7 | 1.7 |
| **Eosinophil count (×10⁹ /L)** | | | | | |
| N | 30 | 9 | 60 | 69 | 99 |
| Mean | 0.0 | 0.0 | 0.0 | 0.0 | 0.0 |
| SD | 0.1 | 0.0 | 0.1 | 0.1 | 0.1 |
| Median | 0.0 | 0.0 | 0.0 | 0.0 | 0.0 |
| Min | 0.0 | 0.0 | 0.0 | 0.0 | 0.0 |
| Max | 0.2 | 0.1 | 0.6 | 0.6 | 0.6 |
| **Basophil count (×10⁹ /L)** | | | | | |
| N | 30 | 9 | 60 | 69 | 99 |
| Mean | 0.0 | 0.0 | 0.0 | 0.0 | 0.0 |
| SD | 0.0 | 0.0 | 0.0 | 0.0 | 0.0 |
| Median | 0.0 | 0.0 | 0.0 | 0.0 | 0.0 |
| Min | 0.0 | 0.0 | 0.0 | 0.0 | 0.0 |
| Max | 0.2 | 0.1 | 0.1 | 0.1 | 0.2 |
| **Procalcitonin count (ng/ml)** | | | | | |
| N | 27 | 8 | 1 | 9 | 36 |

*Table 1 continued on next page*

*Table 1 continued*

| | R-BAC +DA (n=30) | R-BAC (n=9) | CC-BAC (n=60) | T-BAC (n=69) | Total (n=99) |
|---|---|---|---|---|---|
| Mean | 0.3 | 0.3 | 19.3 | 2.4 | 0.8 |
| SD | 0.4 | 0.3 | - | 6.3 | 3.2 |
| Median | 0.1 | 0.2 | 19.3 | 0.2 | 0.2 |
| Min | 0.1 | 0.1 | 19.3 | 0.1 | 0.1 |
| Max | 1.8 | 0.8 | 19.3 | 19.3 | 19.3 |
| D-dimer (ug/L) FEU | | | | | |
| N | 30 | 9 | 15 | 24 | 54 |
| Mean | 885.0 | 909.1 | 1059.3 | 1003.0 | 937.4 |
| SD | 1154.5 | 1054.1 | 1115.0 | 1071.8 | 1109.6 |
| Median | 545.0 | 570.0 | 600.0 | 585.0 | 570.0 |
| Min | 190.0 | 1.9 | 280.0 | 1.9 | 1.9 |
| Max | 6580.0 | 3570.0 | 4460.0 | 4460.0 | 6580.0 |

groups occurred at similar times and comparable frequencies (*Figure 3—figure supplement 1*). In the participants who were randomized into the COVASE trial, excluding the CC-BAC participants the LS mean Ln (CRP) over 7 days follow-up was 3.10 (95% confidence interval [CI] 2.84–3.35) in R-BAC +DA (n=30), and 3.59 (95% CI, 3.13–4.06) in R-BAC (n=9) participants (*Table 2* and *Figure 3A*). Moreover, in the ITT group which included randomized and CC-BAC groups, the LS mean Ln (CRP) over 7 days follow-up was 3.15 (95% confidence interval [CI] 2.87–3.42) in R-BAC +DA (n=30), and 3.55 (95% CI, 3.35–3.75) in T-BAC (n=69) groups (*Table 2* and *Figure 3A*), p=0.01. This indicates a reduction in mean CRP of approximately 33% in the R-BAC +DA group (23.23 mg/mL) compared to the T-BAC group (34.82 mg/mL) at mean follow-up over 7 days. The suppressive effect of dornase alfa on CRP was confirmed in various other subgroup analyses: the per-modified-protocol population only, R-BAC +DA participants, or CC-BAC participants alone (*Table 2*). In addition, to ensure that the CC-BAC participants did not have a significantly different CRP trajectory than the randomized R-BAC control group, we compared the R-BAC to the CC-BAC group by excluding participants randomized to R-BAC +DA and found no significant differences (*Table 2*). Sensitivity analyses supported the observed effect on CRP. These included the Ln(CRP) as an area under the curve; the CC-BAC matched for the last pre-dexamethasone CRP measurement as opposed to their first CRP reading after starting dexamethasone; and the effect of remdesivir or tocilizumab (*Table 2*). A daily mean Ln CRP measurement chart is also provided (*Table 3*).

To better understand the effects of dornase alfa on the CRP concentration in the treated population, we also performed a frequency distribution analysis of the change in CRP over time. Instead of fitting a slope on the CRP values of the individual samples of each participant, we plotted the fraction of the CRP measurment in the final sample collected for each participant during the 7 day treatment period, over the corresponding baseline CRP measurement. We then plotted the distribution of participants within the different ranges of change in CRP (*Figure 3B*). Compared to T-BAC, the R-BAC +DA treatment group exhibited a higher CRP reduction distribution with fewer participants that did not respond to anti-inflammatory therapy as indicated by an increase in CRP concentrations over the course of the 7 day treatment period. Therefore, the addition of dornase alfa to dexamethasone resulted in a significant and sustained reduction in CRP in a greater number of COVID-19-infected participants compared to treatment with dexamethasone alone.

## Secondary outcomes
The length of hospitalisation was analyzed as a time-to-event outcome of alive discharge from the hospital censored at 35 days. The hazard ratio observed in the Cox proportional hazards model was 1.63 (95% CI, 1.01–2.61), p=0.03 (*Figure 3C* and *Figure 3—figure supplement 1*). Therefore,

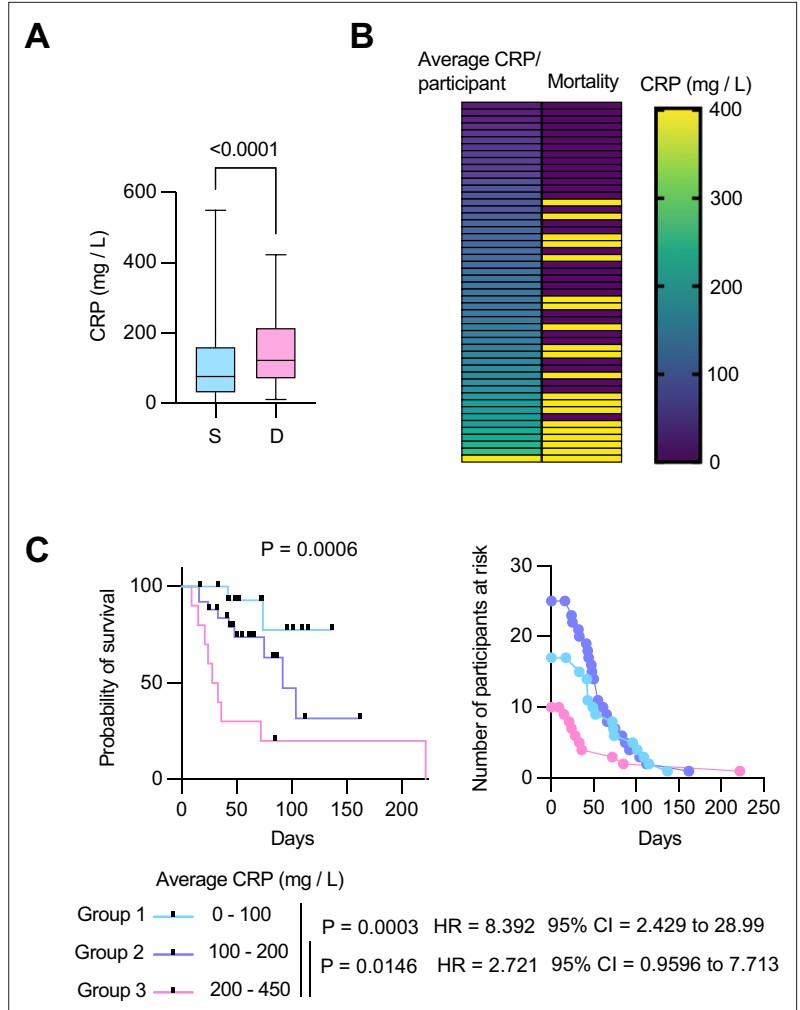

**Figure 2.** Longitudinal C-reactive protein (CRP) predicts survival probability in severe COVID-19 pneumonia. (**A**) Individual CRP concentrations in 465 plasmas from 63 participants with maximum WHO severity grade 7 COVID-19 pneumonia segregated into survivors (n=43) and deceased (n=20) groups from the Berlin COVID-19 study. (**B**) Participants ordered by their longitudinal average CRP concentrations shown in the left column. Mortality is depicted in yellow in the right column. (**C**) Kaplan Meier survival probabilities (left panel) and numbers at risk (right panel) for patients segregated into three categories of longitudinal average CRP ranges: 0–100 mg / L (n=17), 100–200 mg / L (n=25), and 200–450 mg / L (n=10). Statistical significance (P), Hazard ratios (HR) and 95% confidence intervals (95% CI) for group 1 against group 3 and group 2 against group 3 are shown below the survival plot. Statistics by Mann-Whitney and Mantel-Cox log rank tests.

The online version of this article includes the following source data for figure 2:

**Source data 1.** C-reactive protein (CRP) concentrations, participant stratification, and survival analysis of patients in the Berlin COVID-19 study.

throughout the 35 day follow-up period, participants receiving dornase alfa had a 63% higher chance of discharge alive than the T-BAC control group. Although the rate of discharge was similar in 50% of participants, 80% discharge occurred by 8 days in the R-BAC +DA group whereas, whereas the same proportion was reached at 30 days for T-BAC participants. This trend was also observed when only the randomized R-BAC control participants were considered, although not powered to reach significance and with a smaller HR of 1.18 (95% CI, 0.52–2.69), p=0.62, (***Supplementary file 1B*** and ***Figure 3— figure supplement 1***).

Over 7 days of follow up there was no significant difference between the R-BAC +DA and T-BAC groups in either the fraction of participants admitted to ICU (23.3% vs 21.74%), p=0.866, or the length of ICU stay, LS mean 21.25 (95% CI, 4.65–37.84) hr versus 19.85 (95% CI, 8.00–31.70) hr, p=0.883. The

**Table 2.** Primary endpoint and sensitivity analysis.

Mean CRP concentrations over 7 days follow up from baseline for CC-BAC, R-BAC, and R-BAC +DA participants and different comparisons between the three groups as a whole, or stratified by whether in addition to dexamethasone they also received treatment with either remdesivir or tocilizumab at baseline and during the course of the assessment. The difference between the mean log CRP with 95% confidence interval (95% CI) and statistical test values in the group comparisons are shown.

| CRP (mg/L) | BAC+DA | BAC | Difference | p-value* |
|---|---|---|---|---|
| **ITT population (R-BAC+DA, R-BAC, CC-BAC)** | | | | |
| N | 30 | 69 | | |
| LS means log(CRP)* (95% CI) | 3.15 (2.87–3.42) | 3.55 (3.35–3.75) | –0.4 (–0.71 to –0.10) | 0.010 |
| LS means CRP† (95% CI) | 23.23 (17.71–30.46) | 34.82 (28.55–42.47) | 0.67 (0.49–0.91) | |
| **Sensitivity Analyses** | | | | |
| **PP population (R-BAC+DA, R-BAC, CC-BAC)** | | | | |
| N | 29 | 68 | | |
| LS means log(CRP)* (95% CI) | 3.12 (2.85–3.39) | 3.55 (3.36–3.74) | –0.43 (–0.73 to –0.13) | 0.006 |
| LS means CRP† (95% CI) | 22.64 (17.35–29.54) | 34.82 (27.7–42.21) | 0.65 (0.48–0.88) | |
| **ITT population (R-BAC+DA, R-BAC)** | | | | |
| N | 30 | 9 | | |
| LS means log(CRP)* (95% CI) | 3.1 (2.84–3.35) | 3.59 (3.13–4.06) | –0.5 (–0.97 to –0.02) | 0.041 |
| LS means CRP† (95% CI) | 22.12 (17.16–28.5) | 36.34 (22.79–57.94) | 0.61 (0.38–0.98) | |
| **ITT population (R-BAC +DA, CC-BAC)** | | | | |
| N | 30 | 60 | | |
| LS means log(CRP)* (95% CI) | 3.18 (2.91–3.45) | 3.56 (3.35–3.76) | –0.37 (–0.68 to –0.06) | 0.019 |
| LS means CRP† (95% CI) | 24.09 (18.36–31.6) | 35.03 (28.44–43.15) | 0.69 (0.5–0.94) | |
| **ITT population (R-BAC, CC-BAC)** | | | | |
| N | 9 | 60 | | |
| LS means log(CRP)* (95% CI) | 3.78 (3.23–4.33) | 3.53 (3.3–3.77) | 0.24 (−0.32–0.8) | 0.386 |
| LS means CRP† (95% CI) | 43.69 (25.18–75.8) | 34.23 (27.11–43.24) | 1.28 (0.73–2.23) | |
| **Area under the log(CRP), standardized by days followed up, over 7 days follow-up** **ITT population (R-BAC +DA, R-BAC, CC-BAC)** | | | | |
| N | 30 | 69 | | |
| LS means area ‡ (95% CI) | 3.45 (3.22–3.68) | 3.72 (3.55–3.88) | –0.27 (–0.53 to –0.01) | 0.043 |
| **ITT population (R-BAC +DA, R-BAC, CC-BAC) including last pre-dexamethasone CRP value** | | | | |
| N | 30 | 69 | | |
| LS means log(CRP)* (95% CI) | 3.16 (2.83–3.5) | 3.69 (3.44–3.93) | –0.53 (–0.91 to –0.14) | 0.007 |

*Table 2 continued on next page*

*Table 2 continued*

| CRP (mg/L) | BAC+DA | BAC | Difference | p-value* |
|---|---|---|---|---|
| LS means CRP† (95% CI) | 23.57 (16.85–32.970) | 39.92 (31.32–50.89) | 0.59 (0.4–087) | |
| **ITT population (R-BAC +DA, R-BAC, CC-BAC) stratified by BAC treatment** | | | | |
| **No remdesivir or tocilizumab** | | | | |
| N | 12 | 39 | | |
| LS means log(CRP)* (95% CI) | 3.29 (2.83–3.76) | 3.75 (3.45–4.04) | –0.45 (–0.96–0.05) | 0.079 |
| LS means CRP† (95% CI) | 26.97 (16.87–43.11) | 42.35 (31.44–57.04) | 0.64 (0.38–1.06) | |
| **Remdesivir no tocilizumab** | | | | |
| N | 16 | 23 | | |
| LS means log(CRP)* (95% CI) | 3.16 (2.79–3.53) | 3.5 (3.18–3.83) | –0.35 (–0.79–0.1) | 0.123 |
| LS means CRP† (95% CI) | 23.53 (16.29–33.99) | 33.26 (23.97–46.15) | 0.71 (0.45–1.1) | |
| **Tocilizumab no remdesivir** | | | | |
| N | 1 | 5 | | |
| **Remdesivir and tocilizumab** | | | | |
| N | 1 | 2 | | |

*From linear repeated measures model, adjusted for natural log(baseline CRP, age, sex, BMI, serious condition, time, treatment, a treatment*time interaction, and subject as a random effect. Least squares means compared at mean follow-up time).

† Antilog of estimates from *Ratio of BAC +dorna-alfa: BAC shown in the difference column.

‡ From linear model, adjusted for natural log(baseline CRP, age, sex, BMI, serious condition, and treatment).

same trend was observed over the 35 day follow-up period, LS mean 55.21 (95% CI, –23.59–134.00) hr versus 60.60 (95% CI, 4.34–116.86) hr, p=0.905. At any point during the 35 days follow-up, 23% of R-BAC +DA participants were admitted to ICU compared to 23.19% T-BAC participants, p=0.983 (*Supplementary file 1C*). Similarly, there was no significant difference in the time requiring oxygen between the two groups (R-BAC-DA vs T BAC), at either 7 days, LS mean 94.32 (95% CI, 72.8–115.79) hr, versus 88.96 (95% CI, 73.64–104.29) hr, p=0.662, or 35 days, LS mean 133.22 (95% CI, 52.01–214.43) hr versus 156.35 (95% CI, 98.36, 214.33) hr, p=0.618. At 35 days follow up, there were only 9 randomized participants to evaluate, but mean oxygen use exhibited a reduction of 123 hr in R-BAC + DA participants, versus 241 hr for the T-BAC group, p=0.187 (*Supplementary file 1C*).

The time to event data was censored at 28 days after the last dornase alfa dose (up to day 35) for the randomized participants (R-BAC and R-BAC +DA) and at the date of the last electronic record for the CC-BAC group. Over 35 days follow up, one person amongst the 30 patients in the R-BAC +DA group died, compared to 8 of the 69 T-BAC participants. The hazard ratio observed in the Cox proportional hazards model was 0.47 (95% CI, 0.06–3.86), indicating a trend towards a reduced chance of death at any given time-point in R-BAC +DA compared to T-BAC, but this did not reach significance p=0.460 (*Figure 3D* and *Figure 3—figure supplement 1*). The hazard ratio estimates that throughout 35 days follow-up, there was a 53% lower chance of death at any given timepoint in the R-BAC +DA group compared to the T-BAC group. However, the confidence intervals are wide due to the small number of events and consequently the p-value from a log-rank test was 0.460, which does not reach statistical significance.

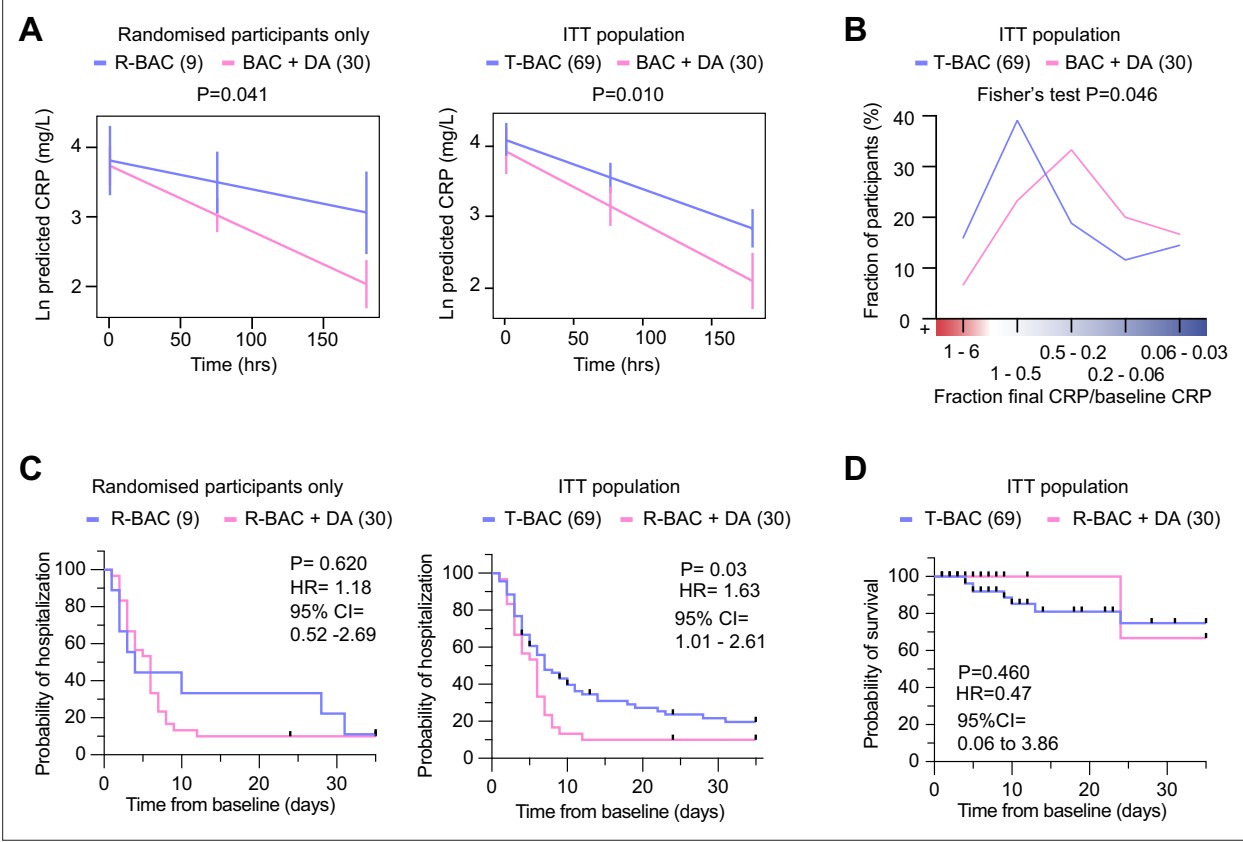

**Figure 3.** Analysis of primary and clinical endpoints. (**A**) Fitted mean (95% confidence interval) from mixed model of natural log (C-reactive protein, CRP) over 7 days follow-up as the outcome. (Left panel) randomized participants only: Blue: participants randomized to R-BAC, n=9; Pink: participants randomized to R-BAC +DA, n=30. (Right panel) ITT population. Blue: T-BAC (CC-BAC and R-BAC) n=69; Pink: R-BAC +DA, n=30. Results were adjusted for natural log baseline CRP, age, sex, BMI, serious comorbidity (diabetes, cardiovascular disease, or hypertension), time and a treatment × time interaction. P-value generated by comparing least-square means between the arms. (**B**) Distribution of participants based on the change in CRP measured as a ratio of the final CRP reading within the 7 day treatment period over the baseline CRP reading per participant. Statistical analysis by Fisher's test. (**C**) Kaplan-Meier plot showing time to discharge from hospital from baseline. Hazard ratio from Cox proportional hazards model adjusted for baseline CRP, age, sex, BMI, serious comorbidity. p-value from log-rank test. Blue: CC-BAC and participants randomized to R-BAC, n=69. Pink: participants randomized to R-BAC +DA, n=30. (**D**) Kaplan-Meier plot showing time to death over 35 days follow up. Hazard ratio from Cox proportional hazards model adjusted for baseline CRP, age, sex, BMI, serious comorbidity. p-value from log-rank test.

The online version of this article includes the following source data and figure supplement(s) for figure 3:

**Source data 1.** Mean C-reactive protein (CRP), participant distribution by CRP change, probability of hospitalization and survival for the randomized and intention-to-treat (ITT) populations.

**Figure supplement 1.** Primary and clinical endpoints.

**Figure supplement 1—source data 1.** Individual C-reactive protein (CRP) readings and numbers at risk for length of hospitalization.

There was no significant difference at either 7- or 35 days follow-up, in the number of participants that required mechanical ventilation in R-BAC +DA compared with T-BAC (16.67% vs 13.04%), p=0.628. Amongst participants that were ventilated, the mean length of mechanical ventilation at 7 days follow-up in R-BAC +DA participants was 76.8 hr, compared to 88.78 hr in the T-BAC group. At 35 days follow-up, the mean length of mechanical ventilation in BAC +DA was 76.8 hr compared to 411.17 hr in T-BAC participants (*Supplementary file 1C*). There was no significant difference in superadded bacterial pneumonia at either 7- or 35 days follow-up: at 7 days, 1 (3.33%) participant in BAC +DA compared to 3 (4.35%) participants in T-BAC, p=0.934; at 35 days, 2 (6.67%) R-BAC +DA participants had bacterial pneumonia, compared to 3 (4.35%) T-BAC participants, p=0.548 (*Supplementary file 1C*).

Blood analysis with no adjustment for multiple testing showed a significant treatment effect in the BAC +DA group against the T-BAC group comparison for three parameters: lymphocyte counts

**Table 3.** Primary endpoint by day.

Mean Log CRP concentrations and standard deviation (SD) for each day over 7 days follow-up by treatment for the intention-to-treat (ITT) population including all individuals (R-BAC +DA, R-BAC, CC-BAC).

| Days from baseline | Mean T-BAC +DA | SD T-BAC +DA | N T- BAC +DA | Mean R-BAC | SD R-BAC | N R-BAC |
|---|---|---|---|---|---|---|
| 1 | 4.058 | 0.591 | 21 | 4.029 | 0.734 | 42 |
| 2 | 3.538 | 0.658 | 33 | 3.852 | 0.937 | 52 |
| 3 | 3.332 | 0.816 | 24 | 3.839 | 1.037 | 53 |
| 4 | 2.917 | 1.017 | 17 | 3.573 | 1.132 | 37 |
| 5 | 2.52 | 1.118 | 15 | 3.872 | 1.157 | 24 |
| 6 | 3.059 | 1.508 | 12 | 3.547 | 1.516 | 26 |
| 7 | 3.386 | 1.824 | 9 | 3.504 | 1.451 | 27 |
| 8 | 3.428 | - | 1 | 2.478 | 1.393 | 9 |

and D-dimer, and procalcitonin (PCT) concentrations. First, R-BAC +DA exhibited higher lymphocyte counts with an LS mean of 0.87 (95% CI, 0.76–0.98) in the T-BAC group vs. 1.08 (95% CI, 0.92–1.27) in R-BAC +DA participants, p=0.02 (*Supplementary file 1B*). Amongst individuals with lymphopenia at baseline (<1 × $10^9$ lymphocytes/L), the R-BAC +DA group exhibited a greater increase in blood lymphocyte numbers compared to the T-BAC group throughout the entire length of treatment (*Figure 4A*). Furthermore, D-dimer levels were lower in R-BAC +DA participants compared to T-BAC participants, with an LS mean D-dimer difference of 1657 (95% CI, 3131–877) (*Figure 4B*, *Supplementary file 1B*). R-BAC +DA participants also exhibited lower PCT concentrations, mean 0.18 ng/mL (95% CI, –0.2–0.56) compared to the T-BAC group, mean 1.31 ng/mL (95% CI, 0.56–2.05), p=0.005 (*Supplementary file 1B*). Repeat analysis excluding the CC-BAC population replicated these results, and changes in all three parameters were significant.

## Exploratory outcomes

Given the potential role of circulating extracellular chromatin in pathology, we examined whether the pulmonary administration of dornase alfa influenced plasma cf-DNA levels. Compared to anonymized healthy donor volunteers at the Francis Crick institute (HD), cf-DNA levels were elevated in the plasmas of all R-BAC and R-BAC-DA randomized COVASE participants. There was no difference in baseline plasma cf-DNA levels between the two randomized participant groups. Notably, during the treatment period plasma cf-DNA was reduced in R-BAC +DA compared to R-BAC participants (*Figure 4C*). Moreover, there was a positive correlation between the levels of cf-DNA in the final sample collected during the treatment period and the magnitude of CRP reduction measured as the ratio of the final over the baseline CRP readings ($CRP_{final}/CRP_{baseline}$) (*Figure 4D*). Moreover, plasma samples with [cf-DNA] >100 µg/mL exhibited significantly higher D-dimer levels compared to samples with [cf-DNA] <100 µg/mL (*Figure 4E*). Hence, dornase alfa treatment was associated with a suppression of circulating cf-DNA that correlated with a reduced risk for coagulopathy and a more sustained reduction in CRP.

## Safety

Dornase alfa was well tolerated with no systemic side-effects which is consistent with its short half-life in vivo (*Supplementary file 1D*). Overall, there were 10 reported adverse events (AEs) in the study, in 9 R-BAC and 30 R-BAC +DA participants (*Supplementary file 1E*). Of these, one was reported by the clinical team as definitely related and one as unlikely to be related to dornase alfa. No treatment-related serious AEs were reported.

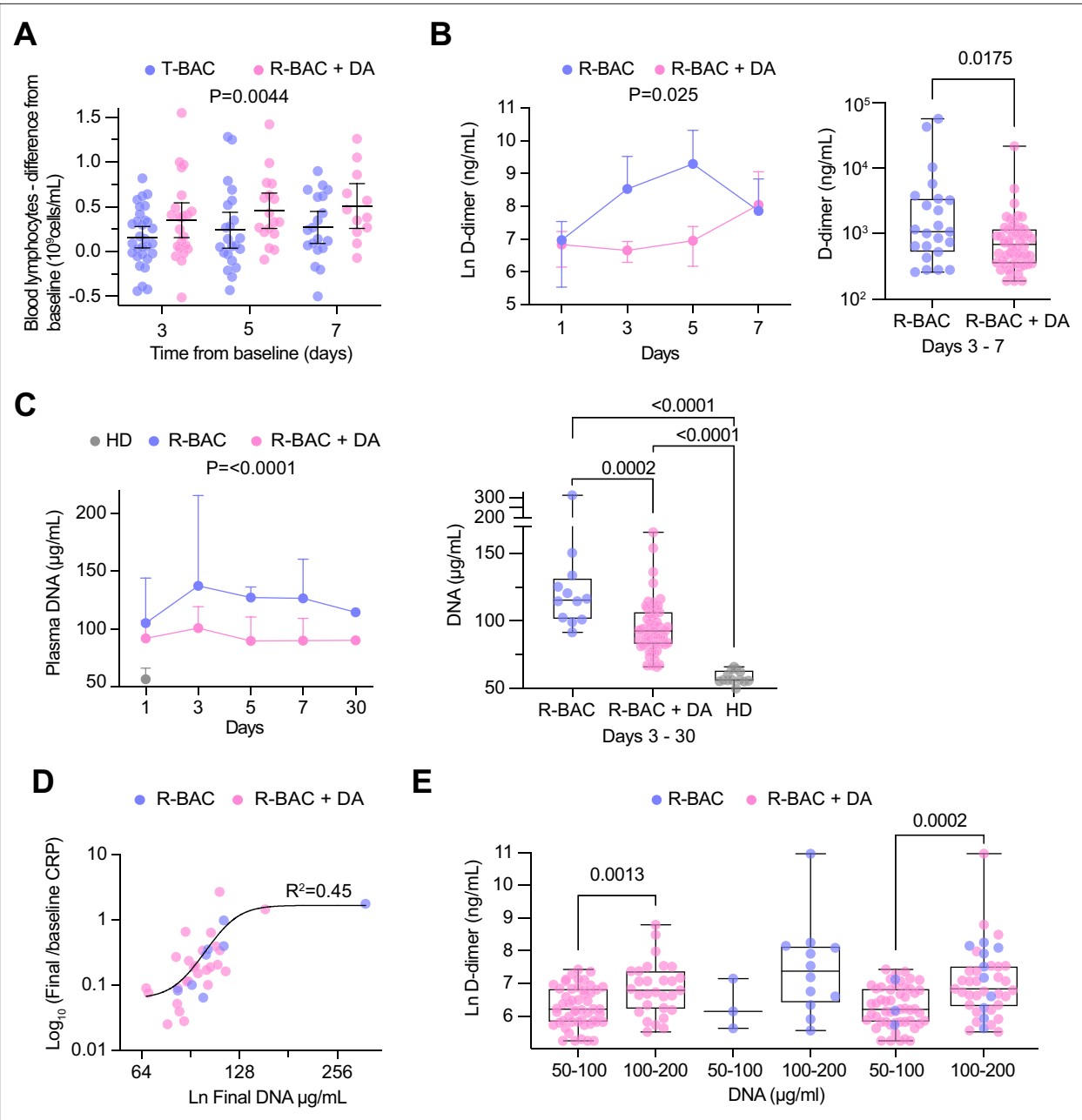

**Figure 4.** Analysis of secondary and exploratory endpoints in blood. (**A**) Difference between the lymphocyte count for each day of the treatment period and the baseline in each intention-to-treat (ITT) participant who exhibited lymphopenia at baseline (<1 × 10⁹ lymphocytes/mL). T-BAC (n=71 samples); R-BAC + DA (n=52 samples). The mean and 95% CI interval are shown with statistical analysis by two-way Anova. (**B**) (Left panel) Mean blood D-dimer levels per day in randomized R-BAC (blue) and R-BAC +DA (pink) participants with error bars depicting 95% CI. Statistical difference by mixed effects Anova analysis. (Right panel) D-dimer concentration in the randomized participant post-baseline blood samples from the R-BAC (n=11 samples) and R-BAC +DA (n=28 samples) groups. Statistical analysis by two-tailed unpaired parametric t-test. (**C**) (Left panel) Mean cell free (cf)-DNA levels per day in randomized R-BAC (n=22 samples, blue) and R-BAC +DA (n=89 samples, pink) participants, with error bars depicting standard deviation. Statistical analysis by mixed effects Anova. (Right panel) Pooled cf-DNA concentration measurements in post-baseline blood samples of R-BAC (n=12) and R-BAC +DA (n=59) groups from days 3, 7 and 30. Healthy donor plasma cf-DNA concentrations (HD, n=13 samples, grey) are shown for comparison. Statistical analysis by one-way Anova. (**D**) Correlation between the final cf-DNA levels and ratio of CRP at day-7 normalized to the baseline C-reactive protein (CRP) (CRP$_{final}$/CRP$_{baseline}$) per randomized participant (Total: n=34; R-BAC: n=7, R-BAC +DA: n=27) . Fitting by non-linear regression. (**E**) Correlation between D-dimer and cf-DNA levels in the blood of participants randomized to R-BAC (blue) or to R-BAC +DA (DA) (pink), where samples were segregated depending on whether the corresponding levels of cf-DNA were <100µg/mL (R-BAC: n=3; R-BAC+ DA: n=51) or >100 µg/mL (R-BAC: n=12; R-BAC+ DA: n=29). Statistical analysis by unpaired parametric t-test.

*Figure 4 continued on next page*

*Figure 4 continued*

The online version of this article includes the following source data for figure 4:

**Source data 1.** Change in blood lymphocytes, D-dimer, DNA and correlations between DNA and D-dimer in randomized participants.

## Discussion

Our findings show that nebulized dornase alfa significantly reduced systemic inflammation as measured by plasma CRP concentrations in patients with severe COVID-19 pneumonia even when already receiving dexamethasone. Dornase alfa exhibited a robust anti-inflammatory effect as assessed by several sensitivity analyses including analysis of only the randomized participant groups, by excluding CC-BAC participants. These findings confirm the usefulness of CRP as a sensitive readout to monitor the impact of anti-inflammatory agents in small cohorts. Improvements in care were also indicated by other biomarkers and secondary endpoints when both randomized and CC-BAC participants were considered. Moreover, despite not being formally powered for length of stay, dornase alfa reduced time to discharge over 35 days.

During the pandemic, the identification of novel and repurposed treatments that would be effective for the treatment of severe COVID-19 pneumonia was hampered by patient recruitment to competing studies. This resulted in small studies with inconclusive or contrary findings. Our study design took advantage of frequent repeated measures of CRP in each patient, to allow a smaller sample size to determine efficacy. CRP was also a primary readout in the CATALYST (*Fisher et al., 2022*) and ATTRACT studies (*Tornling et al., 2021*). In addition, we used contemporary controls as additional comparators to use limited resources more efficiently.

Following the RECOVERY trial, dexamethasone became the standard care in patients with COVID-19 pneumonitis that required oxygen. We recruited participants with CRP ≥30 mg/L, on the day after receiving dexamethasone to minimise steroid-dependent effects on CRP. The finding that dornase alfa can significantly reduce CRP in participants receiving dexamethasone suggests a complementary mechanism of action with significant improvement in existing standard anti-inflammatory care. Moreover, dornase alfa may be a treatment choice for patients with mild COVID-19 pneumonia not requiring oxygen, in whom dexamethasone may be harmful (*Galván-Román et al., 2021*).

The primary endpoint effect was consistently reflected in secondary outcomes. Dornase alfa increased the chance of live discharge by 63% at any time up to 35 days, reducing the overall lenght of hospitalisation. The reduction in hospital occupancy during the COVID-19 pandemic proved critical in sustaining life-saving treatment availability. In addition, dornase alfa significantly increased lymphocyte counts and reduced D-dimer concentrations. In addition to the increased mortality data we provide here, CRP levels are associated with venous thromboembolic disease in COVID-19 with the worst outcomes seen in patients with high CRP and D-dimer concentrations (*Smilowitz et al., 2021*). In a metanalysis of 32 studies involving 10 491 COVID-19 patients, lymphopenia (3.33, p<0.00001), elevated CRP (OR 4.37, p<0.00001), D-dimer (3.39, p<0.00001) and PCT (6.33, p<0.00001) concentrations were independent markers of poor outcomes (*Malik et al., 2021*).

Our study design offered a solution to the early screening of compounds for inclusion in larger platform trials. The study took advantage of frequent repeated measures of quantifiable CRP in each patient, to enable the determination of efficacy using a smaller sample size than if it were powered based on clinical outcomes. We applied a CRP-based approach that was similar to the CATALYST and ATTRACT studies (*Fisher et al., 2022*; *Tornling et al., 2021*). CATALYST showed in comparably small groups (usual care, 54, namilumab, 57 and infliximab, 35) that the GM-CSF-blocking antibody namilumab reduced CRP even in participants treated with dexamethasone, whereas infliximab that targets TNF-α had no significant effect on CRP. This led to a suggestion that namilumab should be considered as an agent to be prioritised for further investigation in the RECOVERY trial. A direct comparison of our results with CATALYST is difficult due to the different nature of the modelling employed in the two studies. Nevertheless, dornase alfa exhibited comparable significance in the reduction in CRP compared to standard of care as described for namilumab at a fraction of the cost. Furthermore, endonuclease therapies may prove safer than cytokine-blocking monotherapies, as they are unlikely to increase the risk of microbial co-infection associated with the neutralization of cytokines that are critical for immune defence such as IL-1β, IL-6 or GM-CSF.

In addition, dornase alfa reduced circulating cf-DNA levels, suggesting that lung administration of the enzyme exerts systemic effects on circulating chromatin. The reduction in plasma cf-DNA levels suggests that by stripping the DNA from chromatin, dornase alfa suppresses the proinflammatory properties of histones and potentiates their degradation (*Papayannopoulos et al., 2011*; *Tsourouktsoglou et al., 2020*). The inverse correlation between the level of circulating cf-DNA and the reduction in CRP during treatment is consistent with the strong functional link between circulating chromatin and systemic inflammation observed in animal models (*Ioannou et al., 2022*). Furthermore, the correlation between D-dimer and cf-DNA, as well as the reduction in D-dimer after dornase alfa treatment, provide functional validation for the suggested pro-thrombotic role of NETs in the alveoli of individuals with severe COVID-19 pneumonia (*Radermecker et al., 2020*). Moreover, the improvement in the recovery rates of lymphopenia support the link between extracellular chromatin and lymphocyte death during sepsis in mice and humans that we also detected in individuals with severe COVID-19 pneumonia (*Aramburu et al., 2022*; *Ioannou et al., 2022*). Another reason why recombinant endonuclease treatment may be warranted in severe COVID-19 and microbial sepsis infections is the significant reduction in endogenous NET chromatin degradation capacity that strongly correlates with a high risk for mortality in these conditions (*Aramburu et al., 2022*).

Nebulized dornase alfa treatment has several advantages. Whilst immunization has reduced COVID-19-related hospital admissions, viral evolution and immune escape may still cause substantial mortality or symptoms associated with post-acute sequelae of SARS-CoV-2 infection (PASC), commonly know as long-COVID. Hence, there will always be a need for virally agnostic therapies that retain efficacy as viruses mutate. Importantly, nebulized dornase alfa has the potential to control immune pathology in a range of infections. Moreover, it can be administered safely and effectively in its nebulized form, outside the health-care setting. Three other small trials of dornase alfa (totalling 18 patients) in COVID-19 have reported improved oxygenation (*Holliday et al., 2021*; *Okur et al., 2020*; *Weber et al., 2020*). One small study indicated improvements in plasma and sputum proteomic profiles (*Fisher et al., 2021*). Despite differing study designs, patient populations and endpoints, there is a consensus for improvement in clinical outcomes.

Yet, there are a few limitations in this study. This single centre open-label study was designed to powerfully report futility or efficacy despite including just under 40 randomized and 60 CC-BAC participants. However, the trial was not powered to report mortality nor to overcome confounders, such as the use of antivirals and tocilizumab, an IL-6 inhibitor recognised to reduce CRP (*Galván-Román et al., 2021*), or the impact of an open-label study on influencing other therapeutic/discharge decisions. Although underpowered, we demonstrate a trend to a reduction in CRP with dornase alfa in participants that had received tocilizumab and/or remdesivir, and those that had not. Moreover, the open-label nature of the study could potentially introduce a placebo effect bias. Nevertheless, we tried to minimize this by applying a standard testing schedule and discharge criteria.

In conclusion, we demonstrate that nebulized dornase alfa significantly reduces inflammation in hospitalised patients with severe COVID-19 pneumonia leading to improved clinical profiles and earlier discharge from hospital. These results confirm the role of extracellular chromatin in promoting systemic inflammation, coagulopathy and immune dysfunction in acute respiratory infections and suggest that recombinant endonucleases provide an additional inexpensive mode of therapeutic intervention that can broaden the effectiveness of anti-inflammatory regimens. These encouraging data warrant further investigation in other serious respiratory infections characterized by hyperinflammation, cell death and NET formation.

## Acknowledgements

We thank the patients, caregivers, and families who participated in the trial; and acknowledge the help of the following: Additional BRC Contributors: Margaret Duku, Gulten Geneci, Farah Islam, Ciprian-Ionut Matei, Marta Merida, Eleni Nastouli, Marivic Ricamara, Anisa Tariq. Pharmacy: Matthew Baker, Nina Bason, Chi Yee Chung, Zoila Gilham-Fernandez, Temi Olusi. Sponsors/UCL: Liam Banks, Helen Cadiou, Novin Fard, Farhat Gilani, Vince Greaves, Yusuf Jaami, Pushpsen Joshi, Misha Ladva, David Lomas, Catherine Maidens, Anthea Mo, Anisha Nayar, Nick McNally, Samim Patel. Data Monitoring Committee: Balaji Ganeshan, Maria Leandro, Kay Roy. COVID Clinical Consultants: Diana Ayoola, Robin Bailey, David Brealey, Mike Brown, Anna Checkley, Charlie Coughlan, Philip Gothard, Robert Heyderman, Sarah Logan, Nicky Longley, Jessica Manson, Michael Marks, David Moore, Neil

Stone, Emma Wall. T8 Nursing Staff: Adam Cureton-Griffiths, Amy Mann, Laura Nichols, Pantelis Savvides. NOCRI Respiratory Translational Research Collaboration: Chris Brightling, Jane Davies, Ratko Djukanovic, Liam Heeney, Ling-Pei Ho, Alex Horsley, Tracy Hussell, Stefan Marciniak, Lorcan McGarvey, Thomas Wilkinson. Pari/Roche Products Limited/LifeArc: Mal Apter, Ruth Davies, Ciara O'Brien, Pauline Stasiak, Davia Viellec. This work was supported by LifeArc (UCL-UCLH132333), UCL, Breathing Matters and the Francis Crick Institute which receives its core funding from the UK Medical Research Council (FC0010129), Cancer Research UK (FC0010129) and the Wellcome Trust (FC0010129). The study was undertaken at UCLH/UCL who received a proportion of funding from the Department of Health's NIHR Biomedical Research Centres funding scheme. VJS and DB are funded by the NIHR University College London Hospitals Biomedical Research Centre. IVA was funded by an EMBO LTF (ALTF 113–2019). Dornase alfa was provided by Roche Products Limited and nebulizers were donated by PARI. Disclosure forms provided by the authors are available with the full text of the published article.

## Additional information

### Competing interests

Jamie Inshaw, Aiden Flynn: Employee of Exploristics. Pauline T Lukey: Employee of Target to Treatment Consulting Ltd. The other authors declare that no competing interests exist.

### Funding

| Funder | Grant reference number | Author |
|---|---|---|
| LifeArc | UCL-UCLH132333 | Joanna C Porter<br>Jamie Inshaw<br>Vincente Joel Solis<br>Emma Denneny<br>Rebecca Evans<br>Mia I Temkin<br>Nathalia De Vasconcelos<br>Iker Valle Aramburu<br>Dennis Hoving<br>Donna Basire<br>Tracey Crissell<br>Jesusa Guinto<br>Alison Webb<br>Hanif Esmail<br>Victoria Johnston<br>Anna Last<br>Thomas Rampling<br>Lena Lippert<br>Elisa Theresa Helbig<br>Florian Kurth<br>Bryan Williams<br>Aiden Flynn<br>Pauline T Lukey<br>Veronique Birault<br>Venizelos Papayannopoulos |
| Francis Crick Institute | FC0010129 | Veronique Birault<br>Venizelos Papayannopoulos<br>Iker Valle Aramburu<br>Mia I Temkin<br>Nathalia De Vasconcelos<br>Dennis Hoving |
| University College London | Breathing Matters | Joanna C Porter |
| NIHR University College London Hospitals Biomedical Research Centre | | Vincente Joel Solis<br>Donna Basire |

| Funder | Grant reference number | Author |
|---|---|---|
| European Molecular Biology Organization | ALTF 113–2019 | Iker Valle Aramburu |

The funders had no role in study design, data collection and interpretation, or the decision to submit the work for publication.

## Author contributions

Joanna C Porter, Resources, Data curation, Supervision, Funding acquisition, Investigation, Methodology, Writing – original draft, Project administration; Jamie Inshaw, Aiden Flynn, Data curation, Formal analysis, Methodology; Vincente Joel Solis, Investigation; Emma Denneny, Rebecca Evans, Donna Basire, Tracey Crissell, Jesusa Guinto, Alison Webb, Hanif Esmail, Victoria Johnston, Anna Last, Thomas Rampling, Bryan Williams, Project administration; Mia I Temkin, Data curation, Formal analysis, Investigation; Nathalia De Vasconcelos, Investigation, Methodology; Iker Valle Aramburu, Dennis Hoving, Data curation, Formal analysis, Investigation, Methodology; Lena Lippert, Elisa Theresa Helbig, Florian Kurth, Data curation, Formal analysis, Investigation, Methodology, Project administration; Pauline T Lukey, Methodology, Project administration; Veronique Birault, Funding acquisition, Project administration; Venizelos Papayannopoulos, Conceptualization, Data curation, Formal analysis, Supervision, Funding acquisition, Investigation, Methodology, Writing – original draft, Project administration, Writing - review and editing

## Author ORCIDs

Victoria Johnston (ID) http://orcid.org/0009-0004-3006-8441
Venizelos Papayannopoulos (ID) http://orcid.org/0000-0002-3741-8190

## Ethics

Clinical trial registration NCT04359654.

The trial was sponsored by University College London (UCL) and carried out at University College London Hospital UCLH with ethical (REC: 20/SC/0197, Protocol: 132333, RAS ID:283091) and UK MHRA approvals. All randomized participants provided informed consent. Consent for CC-BAC participants was covered by Health Service (Control of Participant Information) Regulations 2002. Safety and data integrity were overseen by the Trial Monitoring Group and Data Monitoring Committee. All data was collected at UCLH. Additional CRP data is reported from the Pa-COVID-19 study, Charité Universitätsmedizin Berlin with ethical approval, Berlin (EA2/066/20). Both Pa-COVID-19 and COVASE studies were carried out according to the Declaration of Helsinki and the principles of Good Clinical Practice (ICH 1996). Healthy donor plasma was isolated from anonymized consenting healthy adult volunteers, using protocols approved by the Francis Crick Institute ethics board and in accordance with the Human Tissue Act 2004.

Reviewer #1 (Public review): https://doi.org/10.7554/eLife.87030.4.sa1
Reviewer #2 (Public review): https://doi.org/10.7554/eLife.87030.4.sa2
Author response https://doi.org/10.7554/eLife.87030.4.sa3

# Additional files

## Supplementary files

• Supplementary file 1. Supplementary tables. (A) Dexamethasone administration prior to recruitment Duration of dexamethasone treatment prior to recruitment and initiation of dornase alfa treatment in randomized and contemporary control participants. (B) Secondary endpoints in randomized participants only: Time to discharge, D-dimer, lymphocyte counts and procalcitonin measurements. (C) Secondary clinical endpoints in in randomized and contemporary control participants: Admission to ICU rates, length of stay in ICU, time on oxygen over 7- and 35 days follow-up, duration of mechanical ventilation, and proportion of individuals with superadded bacterial pneumonia. (D) Safety Table depicting the reported adverse events, the degree of severity and the relationship to dornase alfa therapy. (E) Cumulative Summary Tabulations of Serious Adverse Events. Serious adverse effects in randomized R-BAC and R-BAC +DA participants separated by system order class (infections, respiratory, thoracic and mediastinal disorders and vascular disorders).

A total of 6 events were reported, with 2 in the R-BAC and 4 in the R-BAC +DA groups.

• Supplementary file 2. Consort checklist and protocol.

• Supplementary file 3. Statistical analysis plan.

• MDAR checklist

## Data availability

All primary data for every figure presented in the study are included in the accompanying source data files.

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
