## [Editor Report · eLife assessment]

This small-sized clinical trial comparing nebulized dornase-alfa to the best available care in patients hospitalized with COVID-19 pneumonia is **valuable**, but in its present form the paper is **incomplete**.

---

## [Referee Report · Reviewer #1 (Public review)]

This study by Porter et al reports on outcomes from a small, open-label, pilot randomized clinical trial comparing dornase-alfa to best available care in patients hospitalized with COVID-19 pneumonia. As the number of randomized participants is small, investigators describe also a contemporary cohort of controls and the study concludes with decrease of inflammation (reflected by CRP levels) after 7 days of treatment but no other statistically significant clinical benefit.

I read with interest this manuscript and I find the idea about treatment of COVID-19 patients with dornase-alfa novel and inspiring. I have some major concerns about the methodology the authors followed in this RCT.

My major concerns are:

(1) The authors have chosen a primary outcome that cannot be at least considered as clinically relevant or interesting. After 3 years of the pandemic with so much research, why investigate if a drug reduces CRP levels as we already have marketed drugs that provide beneficial clinical outcomes such as dexamethasone, anakinra, tocilizumab and baricitinib.

(2) ΙΤΤ analysis is not followed

---

## [Referee Report · Reviewer #2 (Public review)]

Interesting work with an original and appealing hypothesis. The authors performed an open-label trial comparing nebulized dornase alfa to best available care in COVID-19, reaching the primary outcome of CRP reduction over the first week of intervention. The main weaknesses of the study are the small sample size, the lack of randomization for the majority of the participants, and the lack of blinding. The authors have sufficiently addressed the issues raised, provided that these weaknesses are highlighted in the limitations section.

---

## [Author Response]

The following is the authors’ response to the previous reviews.

Suggestions to the authors:• Please re-analyze findings by omitting from all Tables and Figures all data of comparators who were not randomized (BAC). I understand the difficulties of running this trial but the results of excess reduction of mortality do not allow the publication of a trial where comparators do not come from the randomized patient population.

We wish to thank the editors and reviewers for their useful comments. Given that the study was designed with both randomised and CC participants we can’t easily exclude the CC analysis from the paper. However, we do provide graphs for both randomised only and randomised and CC participants for the primary and secondary endpoints. The fact that the primary endpoint (CRP) results are mirrored in both instances is also informative form a trial design perspective and indicative of the effect of dornase alfa therapy on inflammation being robust enough to yield the same results with small and larger cohorts.

We agree that there are potential drawbacks of using contemporary controls. To address these potential biases we used CC patients recruited at the same time period at single site using the same selection criteria as the randomised group, which minimised potential bias. However, the enrolment and comparison of CRP in CC-BAC participants to concurrent randomised control R-BAC patients indicated that the two groups responded to BAC treatment in the same manner (Table 2, LS means log(CRP) 3.78 vs 3.53, P=0.386), whereas the R-BAC+DA vs R-BAC group comparison yielded significant differences (Table 2, LS means log(CRP) 3.1 vs 3.59, P=0.041). These comparisons mitigate to a large degree these potential problems.

Still, to make easy to distinguish the groups we now use the following unique nomenclature throughout the manuscript which is clearly defined on ln. 111 and state that comparisons of treated participants were performed with both control groups separately and combined.

R-BAC: Randomised BACCC-BAC: Contemporary control BACR-BAC+DA : Randomised BAC+ dornase alfaT-BAC: R-BAC + CC-BAC

In fact, the most important bias in our study, might actually be the placebo effect, given that participants randomised to BAC did not receive a nebulized control substance. We now discuss these points in more detail in the manuscript and modified the title by removing the reference to a randomised trial and clinical outcomes.

• The presentation remains confusing and the manuscript should be critically revised for clarity. There is a repetition of methods (e.g. lines 176-187 repeat 160-175) and redundant results (e.g. Figure S2, Table 3).

We apologise for the repetition. We removed the repeated text in the Exclusion criteria (lines 176-187 in the old manuscript).

Figure S2 is not related to Table 3. Figure S2 depicts baseline characteristics, whereas Table 3 complements the graph in Figure 3A but lists the mean daily value of the primary endpoint as requested by Reviewer 1 in the first round of revision.

At Table 4: the authors should select one method of illustration for lab results, either Table or figure, without repetitions

We agree and have removed Table 4 leaving the graphs instead.

• Regarding inclusion criteria, it is unclear whether high radiological suspicion is sufficient for inclusion or whether PCR based confirmation is required in all instances (differences in wording between lines 153 and 191), and under which oxygen requirements (lines 155 and 192)

We thank the reviewer for pointing this out. Indeed, radiological suspicion was not sufficient and all participants in this study had a positive PCR test as part of their diagnosis prior to inclusion in the study. The entire eligibility section was rewritten to reflect this important point.

• Table 1 should be merged with Table S2 and a better description of cohort baseline severity (P/F, SOFA, APACHE, organ support, number of patients in each point of the WHO severity score) and treatments should be made available.

We thank the reviewer for this suggestion. We have now merged Table 1 and S2 and included WHO ordinal severity information in Table 1, with median, average, SD, min and max values which reflect the participant distribution. Unfortunately, although the additional requested information was recorded, it was not systematically collected for the analysis of the trial and it was not straight forward to compile at this stage.